

# The Met Office Weather Game: investigating how different methods for presenting probabilistic weather forecasts influence decision-making

Elisabeth M. Stephens[1]; David J Spiegelhalter[2]; Ken Mylne[3]; Mark Harrison[3]

[1] School of Archaeology, Geography and Environmental Science, University of Reading, Whiteknights, RG6 6AB
[2] Statistical Laboratory, Centre for Mathematical Sciences, Wilberforce Road, Cambridge, CB3 0WB
[3] Met Office, Fitzroy Road, Exeter, EX1 3PB

*Correspondence to*: Elisabeth M. Stephens (elisabeth.stephens@reading.ac.uk)

**Abstract.** To inform the way probabilistic forecasts would be displayed on their website the UK Met Office ran an online game as a mass participation experiment to highlight the best methods of communicating uncertainty in rainfall and temperature forecasts, and to widen public engagement in uncertainty in weather forecasting. The game used a hypothetical 'ice-cream seller' scenario and a randomised structure to test decision-making ability using different methods of representing uncertainty and to enable participants to experience being 'lucky' or 'unlucky' when the most likely forecast scenario did not occur.

Data were collected on participant age, gender, educational attainment and previous experience of environmental modelling. The large number of participants (n>8000) that played the game has led to the collation of a unique large dataset with which to compare the impact on decision-making ability of different weather forecast presentation formats. This analysis demonstrates that within the game the provision of information regarding forecast uncertainty greatly improved decision-making ability, and did not cause confusion in situations where providing the uncertainty added no further information.

## 1. Introduction

Small errors in observations of the current state of the atmosphere as well as the simplifications required to make a model of the real world lead to uncertainty in the weather forecast. Ensemble modelling techniques use multiple equally likely realisations (ensemble members) of the starting conditions or model itself to estimate the forecast uncertainty. In a statistically reliable ensemble, if 60% of the ensemble members forecast rain, then there is a 60% chance of rain. This ensemble modelling approach has become common place within operational weather forecasting (Roulston et al. 2006), although the information is more typically used by forecasters to infer and then express the level of uncertainty rather than directly communicate it quantitatively to the public.



The Probability of Precipitation (PoP) is perhaps the only exception, with PoP being directly presented
to the US public since 1965 (NRC 2006), although originally derived using statistical techniques rather
than ensemble modelling. Due to long held concerns over public understanding and lack of desire for
PoP forecasts, the UK Met Office only began to present PoP in an online format in late 2011, with the
BBC not including them in its app until 2018 (BBC Media Centre, 2018). However, an experimental
representation of temperature forecast uncertainty was trialled on a now-discontinued section of the Met
Office website called 'Invent'. To move further towards the presentation of weather forecast uncertainty
a mass participation study was planned to highlight the optimal method(s) of presenting temperature and
rainfall probabilities. This study aimed to build on prior studies that have addressed public understanding
of the 'reference class' of PoP (e.g. Gigerenzer et al. 2005; Morss et al. 2008) and decision-making ability
using probabilistic forecasts (e.g. Roulston; Kaplan 2009; Roulston et al. 2006), and to dig deeper into
the conclusions that suggest that there is not a perfect "one size fits all" solution to probabilistic data
provision (Broad et al. 2007).
### 1.1. Public understanding of uncertainty

Numerous studies have assessed how people interpret a Probability of Precipitation (PoP) forecast,
considering whether the PoP reference class is understood, e.g. '10% probability' means that it will rain
on 10% of occasions on which such a forecast is given for a particular area during a particular time period
(Gigerenzer et al. 2005; Handmer; Proudley 2007; Morss et al. 2008; Murphy et al. 1980). Some people
incorrectly interpret to mean that it will rain over 10% of the area or for 10% of the time. Morss et al
(2008) find a level of understanding of around 19% among the wider US population, compared to other
studies finding a good level of understanding in New York (~65%) (Gigerenzer et al. 2005), and 39%
for a small sample of Oregon residents (Murphy et al. 1980). An Australian study found 79% of the
public to choose the correct interpretation, although for weather forecasters (some of whom did not issue
probability forecasts) there is significant ambiguity with only 55% choosing the correct interpretation
(Handmer; Proudley 2007).

The factors which affect understanding are unclear, with Gigerenzer et al. (2005) finding considerable
variation between different cities (Amsterdam, Athens, Berlin, Milan, New York) that could not be
attributed to an individual's length of exposure to probabilistic forecasts. This conclusion is reinforced
by the ambiguity among Australian forecasters, which suggests that any confusion is not necessarily
caused by lack of experience. But as Morss et al. (2008) concluded, it might be more important that the
information can be used in a successful way than understood from a meteorological perspective.
Accordingly, Joslyn et al. (2009) and Gigerenzer et al. (2005) find that decision-making was affected by
whether the respondents could correctly assess the reference class, but it is not clear whether people can
make better decisions using PoP than without it.

Evidence suggests that most people surveyed in the US find PoP forecasts important (Lazo et al. 2009;
Morss et al. 2008), and that the majority (70%) of people surveyed prefer or are willing to receive a





forecast with uncertainty information (with only 7% preferring a deterministic forecast). Research also
suggests that when weather forecasts are presented as deterministic the vast majority of the US public
form their own nondeterministic perceptions of the likely range of weather (Joslyn; Savelli 2010; Morss
et al. 2008). It therefore seems inappropriately disingenuous to present forecasts in anything but a
probabilistic manner, and, given the trend towards communicating PoP forecasts, research should be
carried out to ensure that weather forecast presentation is optimised to improve understanding.
**1.2.  Assessing decision-making under uncertainty in weather forecasting**

Experimental economics has been used as one approach to test decision-making ability under uncertainty,
by incorporating laboratory based experiments with financial incentives. Using this approach, Roulston
et al. (2006) show that, for a group of US students, those that were given information on the standard
error in a temperature forecast performed significantly better than those without. Similarly Roulston and
Kaplan (2009) found that for a group of UK students, on average, those students provided with the 50th
and 90th percentile prediction intervals for the temperature forecast were able to make better decisions
than those who were not. Furthermore, they also showed more skill where correct answers could not be
selected by an assumption of uniform uncertainty over time. This approach provides a useful
quantification of performance, but the methodology is potentially costly when addressing larger numbers
of participants. Criticism of the results has been focused on the problems of drawing conclusions from
studies sampling only students which may not be representative of the wider population; indeed, it is
possible that the outcomes would be different for different socio-demographic groups. However,
experimental economics experiments enable quantification of decision-making ability, and should be
considered for the evaluation of uncertain weather information.

On the other hand, qualitative studies of decision-making are better able to examine in-depth responses
from participants in a more natural setting (Sivle, 2014), with comparability across interviewees possible
by using semi-structured interviews. Taking this approach Sivle (2014) was able to describe influences
external to the forecast information itself that affected a person's evaluation of uncertainty.
**1.3.  Presentation of Uncertainty**

Choosing the format and the level of information content in the uncertainty information is an important
decision, as a different or more detailed representation of probability could lead to better understanding
or total confusion depending on the individual. Morss et al. (2008), testing only non-graphical formats
of presentation, found that the majority of people prefer a percentage (e.g. 10%) or non-numerical text
over relative frequency (e.g. 1 in 10) or odds, but, as noted in the study, user preference does not
necessarily equate with understanding. For complex problems such as communication of health statistics,
research suggests that frequency is better understood than probability (e.g. Gigerenzer et al. 2007), but
for weather forecasts the converse has been found to be true, even when a reference class (e.g. 9 out of



10 computer models predict that …) is included (Joslyn; Nichols 2009). Joslyn and Nichols (2009)
speculate that this response could be caused by the US public's long exposure to the PoP forecast, or
because weather situations do not lend themselves well to presentation using the frequency approach
because unlike for health risks they do not relate to some kind of population (e.g. 4 in 10 people at risk
of heart disease).
As well as assessing the decision-making ability using a PoP forecast, it is also important to look at
potential methods for improving its communication. Joslyn et al. (2009) assess whether specifying the
probability of no rain or including visual representations of uncertainty (a bar and a pie icon) can improve
understanding. They found that including the chance of no rain significantly lowered the number of
individuals that made reference class errors. There was also some improvement when the pie icon was
added to the probability, which they suggested might subtly help to represent the chance of no rain. They
conclude that given the wide use of icons in the media more research and testing should be carried out
on the potential for visualisation as a tool for successful communication.
Tak, Toet and Erp (2015) considered public understanding of 7 different visual representations of
uncertainty in temperature forecasts among 140 participants. All of these representations were some form
of a line chart / fan chart. Participants were asked to estimate the probability of a temperature being
exceeded from different visualisations, using a slider on a continuous scale. They found systematic biases
in the data, with an optimistic interpretation of the weather forecast, but were not able to find a clear
'best' visualisation type.
**2. Objectives and Methodology**
This study aims to address two concerns often vocalised by weather forecast providers about presenting
forecast uncertainties to the public; firstly, that the public do not understand uncertainty; and secondly,
that the information is too complex to communicate. Our aim was to build on the previous research of
Roulston and Kaplan (2009) and Roulston et al. (2006) by assessing the ability of a wider audience (not
only students) to make decisions when presented with probabilistic weather forecasts. Further, we aimed
to identify the most effective formats for communicating weather forecast uncertainty by testing different
visualisation methods and different complexities of uncertainty information contained within them (e.g.
a descriptive probability rating (Low (0%-20%), Medium (30%-60%) or High (70%-100%) compared to
the numerical value).
As such our objectives are as follows:
• To assess whether providing information on uncertainty leads to confusion compared to a
traditional (deterministic) forecast



• To evaluate whether participants can make better decisions when provided with probabilistic
151       rather than deterministic forecast information


• To understand how the detail of uncertainty information and the method of presenting it might
154       influence this decision-making ability


Socio-demographic information was collected from each participant, primarily to provide information
about the sample, but also to potentially allow for future study of demographic influences.

For this study we focused on two aspects of the weather forecast; precipitation, as Lazo et al. (2009)
found this to be of the most interest to users and PoP has been presented for a number of years (outside
the UK); and temperature, since a part of the UK Met Office website at that time included an indication
of predicted temperature uncertainty ('Invent').

Seven different temperature forecast presentation formats were tested (Fig. 1), representing 3 levels of
information content (deterministic, mean with 5th / 95th percentile range, mean with 5th / 95th and 25th /
75th. These included table and line presentation formats as well as the 'Invent' style as it appeared on the
web, and a more simplified version based on some user feedback. Nine different rainfall forecast
presentation formats were tested (Fig. 2), with 3 different levels of information content including one
deterministic format used as a control from which to draw comparisons. While there are limitless
potential ways of displaying the probability of precipitation, we felt it important to keep the differences
in presentation style and information content to a minimum in order to quantify directly the effect of
these differences rather than aspects like colour or typeface, and so maintain control on the conclusions
we are able to draw.

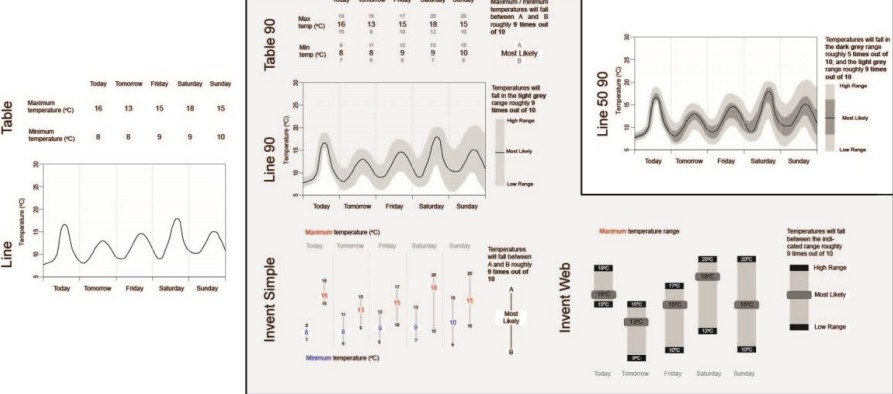



Figure 1: Temperature forecast presentation formats. Two different deterministic formats used for
comparison (a table and a line graph); four different ways of presenting the 5th and 95th percentiles (Table



90, Line 90, Invent Simple, Invent Web; and, a more complex fanchart (Line 50 90) representing, the
$25^{th}$ and $75^{th}$ percentiles as well as the $5^{th}$ and $95^{th}$ shown in Line 90.


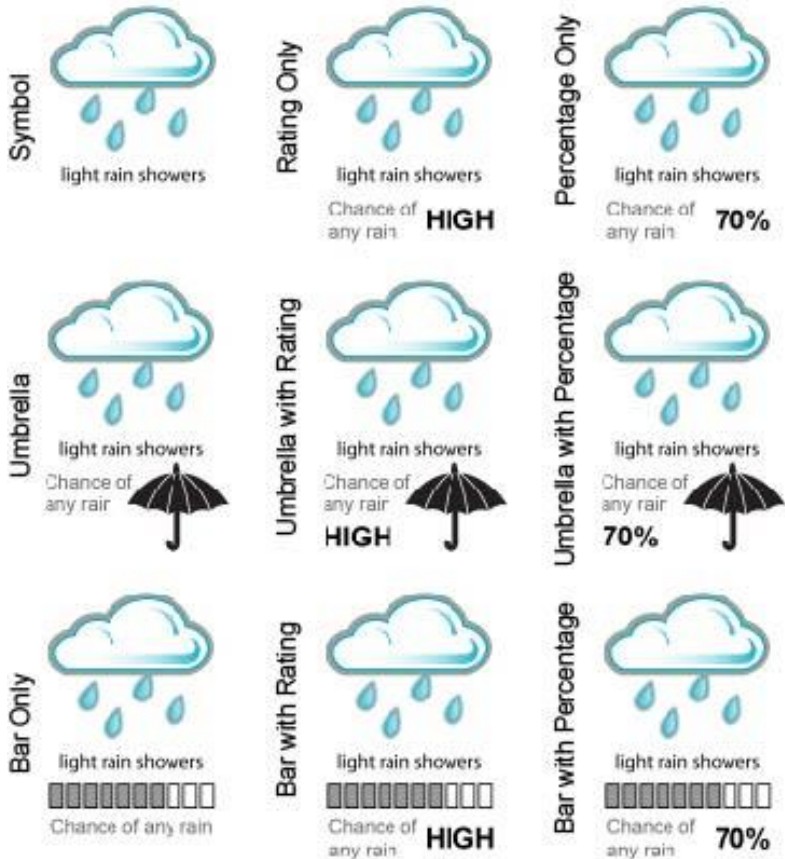


Figure 2: Precipitation presentation formats, with varying levels of information content. Rating is either
Low (0%-20%), Medium (30%-60%) or High (70%-100%), and the Percentage is to the nearest 10%.

Our method of collecting data for this study was an online game linked to a database. Alternative
communication formats can be evaluated in terms of their impacts on cognition (comprehension), affect
(preference) and behaviour (decision- making) impacts. Unpublished focus groups held by the Met
Office had concentrated on user preference, but we chose to focus on comprehension and decision-
making. While previous laboratory-based studies had also looked at decision-making, we hoped that by
using a game we would maximise participation by making it more enjoyable, therefore providing a large
enough sample size for each presentation format to have confidence in the validity of our conclusions.
Since the game was to be launched and run in the UK summer it was decided to make the theme
appropriate to that time of year, as well as engaging to the widest demographic possible. Accordingly,



the choice was made to base the game around running an ice cream van business. The participants would
try to help the ice cream seller, 'Brad', earn money by making decisions based on the weather forecasts.

It is not possible to definitively address all questions in a single piece of work (Morss et al. 2008), and
consequently we focussed on a participant's ability to understand and make use of the presentation
formats. This study does not look at how participants might use this information in a real-life context, as
this would involve other factors such as the 'experiential' as well as bringing into play participants' own
thresholds / sensitivities for risk. By keeping the decisions specific to a theoretical situation (e.g. by using
made-up locations) we hoped to be able to eliminate these factors and focus on the ability to understand
the uncertainty information.

As addressed in Morss et al. (2010), there are advantages and disadvantages with using a survey rather
than a laboratory based experiment, and accordingly there are similar pros and cons to an online game.
In laboratory studies participants can receive real monetary incentives related to their decisions (see
Roulston; Kaplan 2009; Roulston et al. 2006), whereas for surveys this is likely not possible. Our solution
was to make the game as competitive as possible, while being able to identify and eliminate results from
participants who played repeatedly to maximise their score. We also provided the incentive of the
potential of a small prize to those that played all the way to the end of the game.

Surveys are advantageous in that they can employ targeted sampling to have participants that are
representative of the general population, something that might be difficult or cost-prohibitive on a large
scale for laboratory studies. By using an online game format, we hoped to achieve a wide enough
participation to enable us to segment the population by demographics. We thought that this would be
perceived as more fun than a survey and therefore more people would be inclined to play, as well as
enabling us to use social media to promote the game and target particular demographic groups where
necessary. The drawback of an online game might be that it is still more difficult to achieve the desired
number of people in particular socio-demographic groups than if using a targeted survey.

223       **2.1.  Game Structure**


The information in this section provides a brief guide to the structure of the game; screenshots of the
actual game can be found in the electronic supplement.

227       **2.1.1. Demographic Questions, Ethics and Data Protection**


As a Met Office – led project there was no formal ethics approval process, but the ethics of the game
were a consideration and its design was approved by individuals within the Met Office alongside Data
Protection considerations. It was decided that although basic demographic questions were required to be
able to understand the sample of the population participating in the game, no questions would be asked




which could identify an individual. Participants could enter their email address so that they could be
contacted if they won a prize (participants under 16 were required to check a box to confirm they had
permission from a parent or guardian before sharing their email address), however these emails were
kept separate from the game database that was provided to the research team.

On the 'landing page' of the game the logos of the Met Office, University of Bristol (where the lead
author was based at the time) and the University of Cambridge were clearly displayed, and participants
were told that "Playing this game will helps us to find out the best way of communicating the confidence
in our weather forecasts to you", with a 'More Info' taking them to a webpage telling them more about
the study. On the first 'Sign up' page participants were told (in bold font) that "all information will stay
anonymous and private", with a link to the Privacy Policy.

The start of the game asked some basic demographic questions of the participants; age, gender, location
(first half of postcode only) and educational attainment (see supplementary material), as well as two
questions designed to identify those familiar with environmental modelling concepts or aware that they
regularly make decisions based on risk:

Have you ever been taught or learnt about how scientists use computers to model the environment? (Yes,
No, I'm not sure)

Do you often make decisions or judgements based on risk, chance or probability?
(Yes, No, I'm not sure)

The number of demographic questions was kept to a minimum to maximise the number of participants
that wanted to play the game. Following these preliminary questions the participant was directed
immediately to the first round of game questions.
**2.1.2. Game Questions**

Each participant played through four 'weeks' (rounds) of questions, where each week asked the same
temperature and rainfall questions, but with a different forecast situation. The order that specific
questions were provided to participants in each round was randomised to eliminate learning effects from
the analysis. The first half of each question was designed to assess a participant's ability to decide
whether one location (temperature questions) or time period (rainfall questions) had a higher probability
than another, and the second half asked them to decide on how sure they were that the event would occur.
Participants were presented with 11 satellite buttons (to represent 0 to 100%, these buttons initially
appeared as unselected so as not to bias choice) from which to choose their confidence in the event
occurring. This format is similar to the slider on a continuous scale used by Tak, Toet and Erp (2015).

Temperature questions (Fig. 4) took the form:

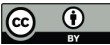




Which town is more likely to reach 20ºC on Saturday? [Check box under chosen location]
How sure are you that it will reach 20ºC here on Saturday? [Choose from 11 satellite buttons on scale
from 'certain it will not reach 20ºC to 'certain it will reach 20ºC']

Rainfall questions (Fig. 5) took the form:

Pick the three shifts where you think it is least likely to rain
How sure are you that it won't rain in each of these shifts?
[Choose from 11 satellite buttons on scale from 'certain it will not rain' to 'certain it will rain']
**2.1.3. Game Scoring and feedback**

The outcome to each question was generated 'on the fly' based on the probabilities defined from that
question's weather forecast (and assuming a statistically reliable forecast). For example, if the forecast
was for an 80% chance of rain, 8 out of 10 participants would have a rain outcome, 2 out of 10 would
not. Participants were scored (S) based on their specified confidence rating (C) and the outcome, using
an adjustment of the Brier Score (BS) (see Table 1), so that if they were more confident they had more
to gain, but also more to lose. So if the participants states a probability of 0.5 and it does rain the BS=0.75
and S=0; if the probability stated is 0.8 and it does rain the BS=0.96 and S=21; if the probability stated
is 0.8 and it doesn't rain the BS= 0.36 and S= –39.




| | $E^0$ | | | | | 50/50 | | | | | $E^1$ |
|---|---|---|---|---|---|---|---|---|---|---|---|
| C | 0 | 0.1 | 0.2 | 0.3 | 0.4 | 0.5 | 0.6 | 0.7 | 0.8 | 0.9 | 1 |
| $S^1$ | -75 | -56 | -39 | -24 | -11 | 0 | 9 | 16 | 21 | 24 | 25 |
| $S^0$ | 25 | 24 | 21 | 16 | 9 | 0 | -11 | -24 | -39 | -56 | -75 |


Table 1: Game scoring based on an adjustment (1) of the Brier Score (BS) (2), where C is the confidence
rating, E is the expected event and S the score for the actual outcome (x), where x=1 if the event occurs
and x=0 if it does not.

$$S^x = 100(BS - 0.75)$$

302 (1)

$$BS = 1 - (x - C)^2$$

304 (2)

This scoring method was chosen as we wanted participants to experience being unlucky, i.e. that they
made the right decision but the lower probability outcome occurred. This meant that they would not





necessarily receive a score that matched their decision-making ability, although if they were to play
through enough rounds then on average those that chose the correct probability would achieve the best
score.
For a participant to understand when they were just 'unlucky', we felt it important to provide some kind
of feedback as to whether they had made the correct decision. It was decided to give players traffic light
coloured feedback corresponding to whether they had been correct [green], correct but unlucky [amber],
incorrect but lucky [amber], or incorrect [red]. The exact wording of these feedback messages was the
subject of much debate. Many of those involved in the development of the weather game who have had
experience communicating directly to the public were concerned about the unintended consequences of
using words such as 'lucky' and 'unlucky'; for example that it could be misinterpreted that there is an
element of luck in the forecasting process itself, rather than the individual being 'lucky' or 'unlucky'
with the outcome. As a result the consensus was to use messaging such as "You provided good advice,
but on this occasion it rained".
**2.2. Assessing participants**
Using the data collected from the game, it is possible to assess whether participants made the correct
decision (for the first part of each question) and how close they come to specifying the correct confidence
(for the second part of each question). For the confidence question we remove the influence of the
outcome on the result by assessing the participant's ability to rate the probability compared to the 'actual'
probability. The participant was asked for the confidence for the choice that they made in the first half
of the question, so not all participants would have been tasked with interpreting the same probability.
**3. Results**
**3.1. Participation**
Using traditional media routes and social media to promote the game we were able to attract 8220 unique
participants to play the game through to the end, with 11398 total plays because of repeat players. The
demographic of these participants was broadly typical of the Met Office web site, with a slightly older
audience, with higher educational attainment, than the wider internet might attract (see Fig. 3).
Nevertheless, there were still over 300 people in the smallest age category (under 16s) and nearly 500
people with no formal qualifications.



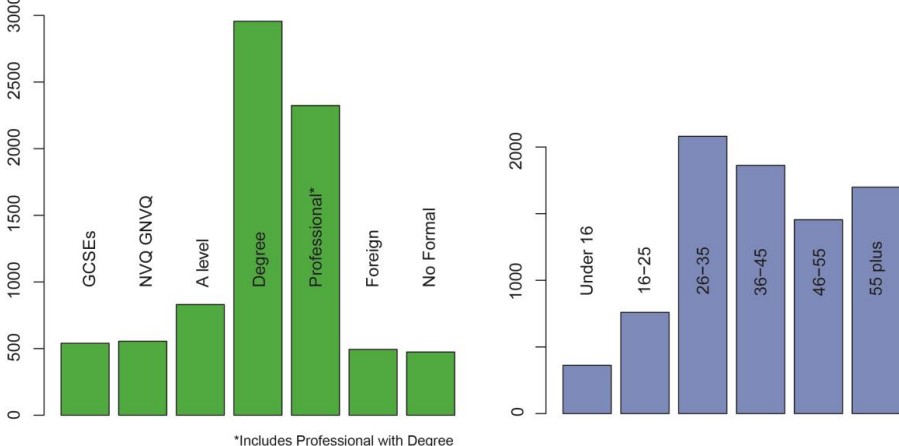



Figure 3: Educational attainment and age structure of participants, full description of educational
attainment in supplementary material.
**3.2.  Assessing participant outcomes**
Before plotting the outcomes we removed repeat players and participants who indicated that they had
been taught or had learnt about environmental modelling, leaving 4686 participants in total. This was so
that we did not bias our analysis by including too many of our peers in meteorology and academia. It
should be noted that for the confidence questions we found that many people specified the opposite
probability, perhaps misreading the question and thinking that it referred to the chance of 'no rain' rather
than 'any rain' as the question specified. We estimate that approximately 15% of participants had this
misconception, although this figure might vary for different demographic groups: it is difficult to specify
the exact figure since errors in understanding of probability would also exhibit a similar footprint in the
results.

For the first part of the temperature and rainfall questions the percentage of participants who make the
correct decision (location choice or shift choice) is calculated. In Fig. 4 and Fig. 5 bar plots present the
proportion of participants who get the question correct, and error bars have been determined from the
Standard Error of the proportion ($SE_p$) (Equation 3). In Figs. 6a and 7a bar plots have been used to present
the mean proportion of the four questions that each participant answers correctly, and error bars have
been determined from the Standard Error of the sample mean (Equation 4). The boxplots in Figs. 6b and
7b include notches that represent the 95% confidence interval around the median.

$$SE_p = \sqrt{p(1-p)n}$$

365    (3)




$$SE_{\bar{x}} = \frac{\sigma}{\sqrt{n}}$$


369 (4)

### 3.3. Results from temperature questions

Figure 4a shows the forecasts presented in the temperature questions for each of the 4 questions (Weeks),
Figure 4b presents the percentage of correct responses for the choice in the first part of the question for
each presentation format, and the Figure 4c presents the error between the actual and chosen probability
in the location chosen for each presentation format.

The scenario in Question 1 was constructed so that it was possible to make the correct choice regardless
of the presentation format. The results show that the vast majority of participants presented with each
presentation format correctly chose Stonemouth as the location where it was most likely to reach 20ºC.
There was little difference between the presentation formats, though more participants presented with
the Line format made the correct choice than for the Table format, despite them both having the same
information content. Participants with all presentation formats had the same median probability error if
they correctly chose Stonemouth. Small sample sizes for Rockford (fewer people answered the first
question incorrectly) limits comparison for those results, as shown by the large notch sizes.

The scenario in Question 2 was for a low probability of reaching 20ºC, with only participants provided
with presentation formats that gave uncertainty information able to see the difference between the two
uncertainty ranges and determine Rockford as the correct answer. The results show that most participants
correctly chose Rockford regardless of the presentation format. In this case the Line format led to poorer
decisions than the Table format on average, despite participants being provided with the same
information content. Invent Web, Invent Simple and Line 90 were the best presentation formats for the
first part of Question 2. For Rockford in the second part of the question only participants given the Table
presentation format had a median error of 0, with other formats leading to an overestimation compared
to the true probability of 30%. Those presented with Line 50 90 who interpreted the graph accurately
would have estimated a probability of around 25%, however the results are no different from the other
presentation formats which present the 5th to 95th percentiles, suggesting that participants were not able
to make use of this additional information.

Question 3 was similar to Question 2 in that only participants provided with presentation formats that
gave uncertainty information were able to determine the correct answer (Stoneford), but in this scenario
the probability of reaching 20ºC is high in both locations. Fewer participants were able to select the
correct location than in Question 2. However, fewer than 50% (getting it right by chance) of those
presented with the Table or Line answered correctly, showing that they were perhaps more influenced
by the forecast for other days (e.g. 'tomorrow' had higher temperature for Stoneford) than the forecast



for the day itself. For the scenario in this question fewer participants with the Line 50 90 format answered
the question correctly than other formats that provided uncertainty information. Despite this, all those
that answered the location choice correctly did fairly well at estimating the probability; the median
response was for a 90% rather than 100% probability which is understandable given that they were not
provided with the full distribution, only the 5[th] to 95[th] percentiles. Despite getting the location choice
wrong, those with Line 90 or Line 50 90 estimated the probability just as well as their counterparts who
answered the location choice correctly.

The location choice in Question 4 was designed with a skew to the middle 50% of the distribution so that
only those given the Line 50 90 presentation format would be able to identify Stoneford correctly; results
show that around 70% of participants with that format were able to make use of it. As expected, those
without this format made the wrong choice of location, and given that the percentage choosing the correct
location was less than 50% (getting it right by chance) it suggests that the forecast for other days may
have influenced their choice (e.g. 'Friday' had higher temperatures in Rockford). Participants with Line
50 90 who made the correct location choice were better able to estimate the true probability than those
who answered the first half of the question incorrectly. Participants without Line 50 90 who answered
the location choice correctly as Stoneford on average underestimated the actual probability; this is
expected given that they did not receive information that showed the skew in the distribution; the
converse being true for 'Rockford'.
### 3.4.  Results from rainfall questions
Figure 5a shows the forecasts presented in the rainfall questions for each of the 4 questions (shifts),
Figure 5b presents the percentage of correct responses for the choice in the first part of the question for
each presentation format, and the Figure 5c presents the error between the actual and chosen probability
in the shifts chosen for each presentation format.

Question 1 was designed so that participants were able to correctly identify the shifts with the lowest
chance of rain (Shifts 2, 3 and 4) regardless of the presentation format they were given. Accordingly the
results for the shift choice show that there is no difference in terms of presentation format. For the
probability estimation Shift 1 can be ignored due to the small sample sizes, as shown by the large notches.
For Shift 2 the median error in probability estimation was 0 for any presentation format which gave a
numerical representation. Those given the risk rating ('medium') overestimated the true chance of rain
(30%) in Shift 2, were correct (though with a higher range of errors) in Shift 3 ('low', 10%), and
overestimated it in Shift 4 ('low', 0%), showing that risk ratings are ambiguous.

Question 2 was set-up so that participants could only identify the correct shifts (Shifts 1, 2 and 3) if they
were given a numerical representation of uncertainty; the difference in probability between Shifts 3
('medium', 40%) and 4 ('medium', 50%) cannot be identified from the rating alone. The results (Figure
5b, Q2) confirmed that those with numerical representations were better able to make use of this



information, though "Bar with Rating" showed fewer lower correct answers. Despite this, over 80% of
those with the deterministic forecast, or with just the rating, answered the question correctly. This
suggests an interpretation based on a developed understanding of weather; the forecasted situation looks
like a transition from dryer to wetter weather. For the probability estimation participants with
presentation formats with a numerical representation did best across all shifts, with the results for "Bar
with Perc" giving the smallest distribution in errors.

Question 3 presented a scenario whereby the correct decision (Shifts 1, 2 and 4) could only be made with
the numerical representation of probability, and not a developed understanding of weather. Consequently
the results show a clear difference between the presentation formats which gave the numerical
representation compared to those that did not, though again "Bar with Rating" showed fewer correct
answers. The results also show that participants provided with the probability rating do not perform
significantly differently from those with the symbol alone, perhaps suggesting that the weather symbol
alone is enough to get a rough idea of the likelihood of rain. For this question the percentage on its own
led to a lower range of errors in probability estimation than "Bar with Perc", as found for Question 2.

The scenario in Question 4 was designed to test the influence of the weather symbol itself by
incorporating two different types of rain; 'drizzle' ('high',90%) and 'heavy rain showers' ('high', 70%).
Far fewer participants answered correctly (Shifts 1, 2 and 3) when provided with only the rating or
symbol, showing that when not provided with the probability information they think the 'heavier' rain is
more likely. This appears to hold true for those given the probability information too, given that fewer
participants answered correctly than in Question 2. This seemed to lead to more errors in the probability
estimation too, with all presentation formats underestimating the probability of rain for 'drizzle' (though
only those who answered incorrectly in the first part of the question would have estimated the probability
for Shift 4).





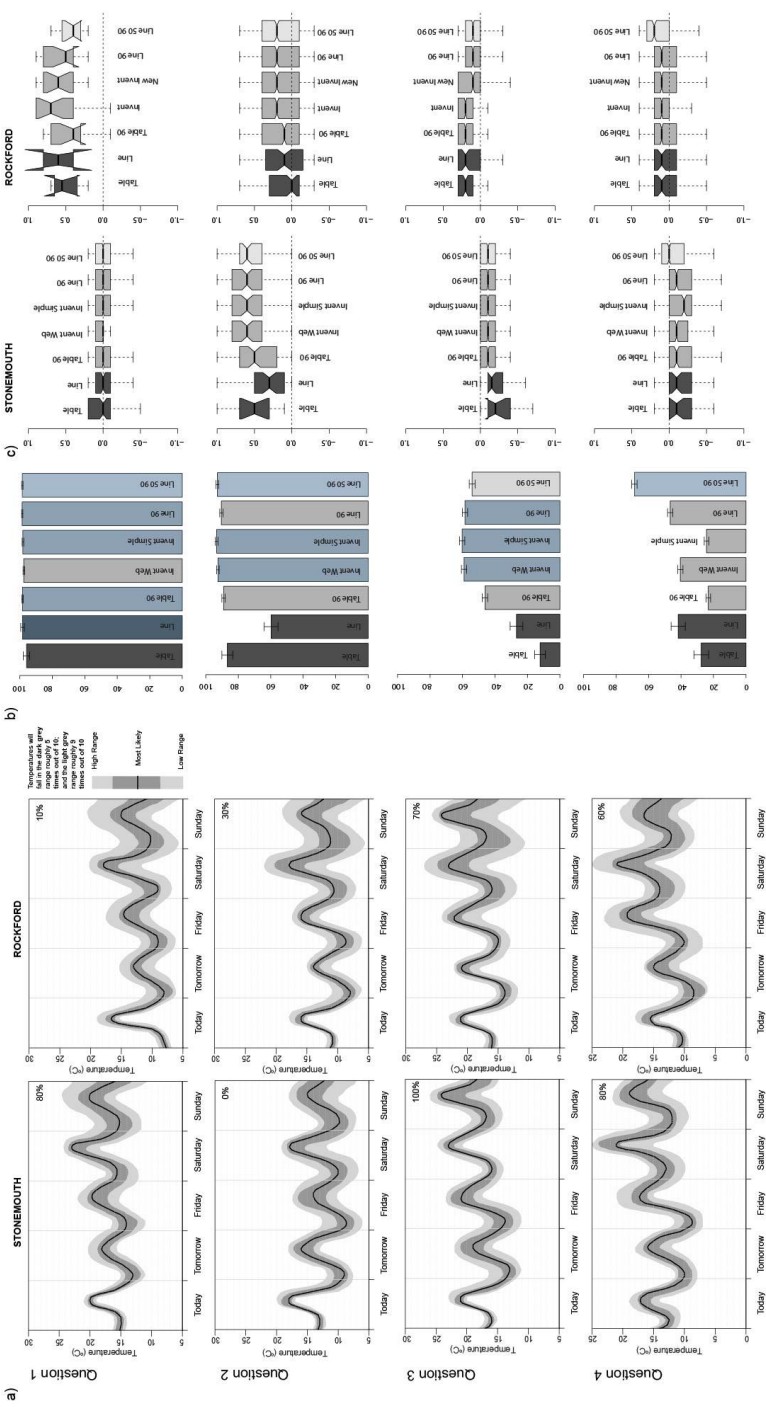

Figure 4: a) temperature questions presented to each participant (the format shown is 'line 50 90'); b) percentage of correct answers for the location choice (blue shading indicates the 'best' performing format); and, c) mean error between stated and actual probability.

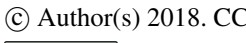


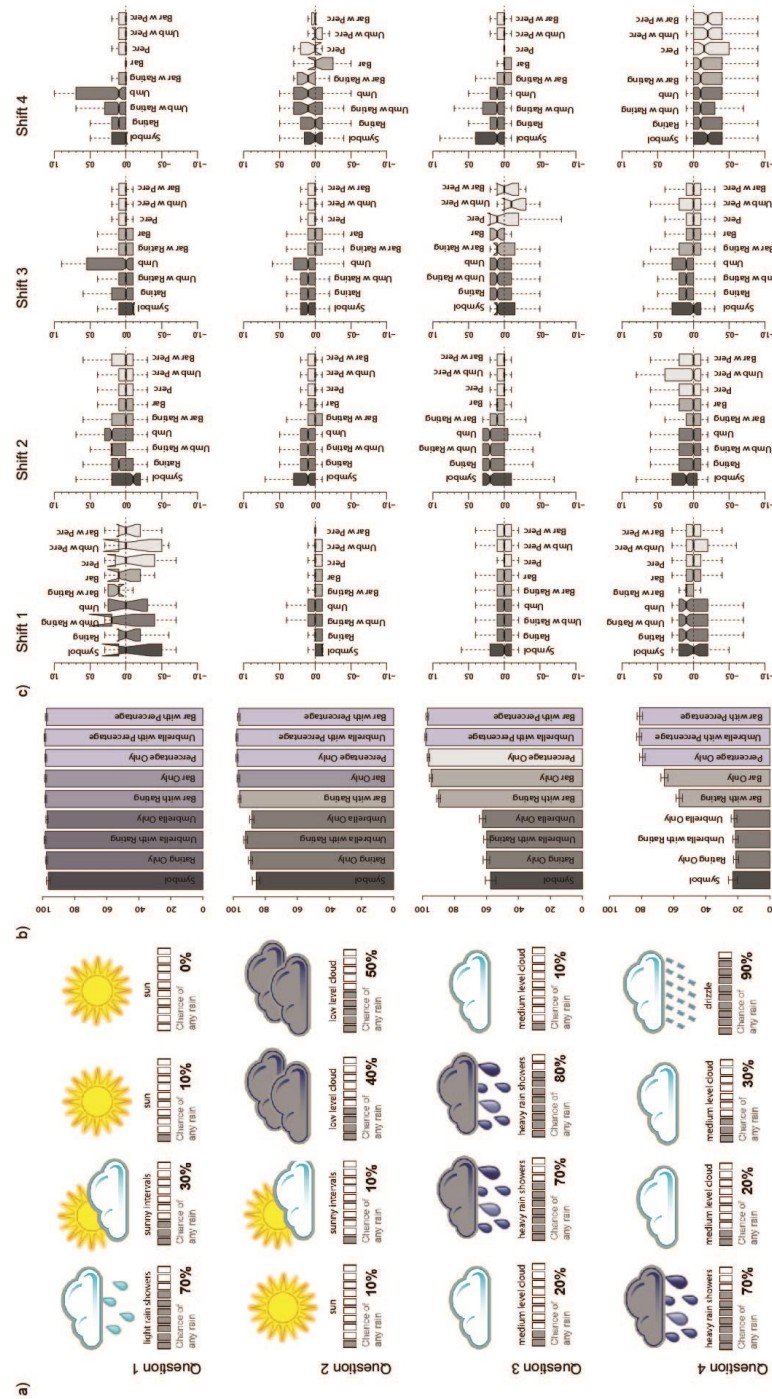

Figure 5: a) Rainfall questions presented to each participant (the format shown is 'Bar with Percentage');
b) percentage of correct answers for the shift choice (blue shading indicates the 'best' performing
format); and, c) mean error between chosen and actual probability.



## 4. Discussion

### 4.1. Does providing information on uncertainty lead to confusion?

We set up Question 1 (Q1) for both the temperature and rainfall questions as a control by providing all participants with enough information to make the correct location / shift choice regardless of the presentation format that they were assigned. The similarity in the proportion of people getting the answer correct for each presentation format in this question (Fig. 4 and 5) demonstrates that providing additional information on the uncertainty in the forecast does not lead to any confusion compared to deterministic presentation formats. Given the small sample size when using subgroups of subgroups, we cannot conclude with any confidence whether age or educational attainment are significant influences on potential confusion.

Previous work has shown that the public infer uncertainty when a deterministic forecast is provided (Joslyn and Savelli, 2010; Morss et al. 2008). Our results are no different; looking in detail at the deterministic 'symbol only' representation for Q1 of the rainfall questions (a 'sun' symbol forecast), 36% of participants indicated some level of uncertainty (i.e. they did not specify the correct value of 0% or misread the question and specify 100%). This shows that a third of people place their own perception of uncertainty around the deterministic forecast. Where the forecast is for 'light cloud' rather than 'sun' this figure goes up to 86%. Similarly for Q1 of the temperature questions, even when the line or the table states (deterministically) that the temperature will be above 20 degrees, the confidence responses for those presentation formats shows that the median confidence from participants is an 80% chance of that temperature being reached.

### 4.2. What is the best presentation format for the Probability of Precipitation?

The amount of uncertainty that participants infer around the forecast was examined by looking at responses for a shift where a 0% chance of rain is forecast (see Fig. 5, Q1, shift 4). For this question, participants were given a 'sun' weather symbol, and / or a 'low' rating or 0% probability. The presentation format that leads to the largest number of precise interpretations of the actual probability is 'bar only', but the results are similar for any of the formats that provide some explicit representation of the probability.

Participants that were assigned formats that specified the probability rating (High / Med / Low) gave fewer correct answers, presumably because they were told that there was a 'low' rather than 'no' chance of rain. Arguably this is a positive result, since it indicates that participants take into account the additional information and are not just informed by the weather symbol. However, it also highlights the potential problem of being vague when forecasters are able to provide more precision. Providing a probability rating could limit the forecaster when there is a very small probability of rain; specifying a




rating of 'low' is perhaps too vague, and specifying 'no chance' is more akin to a deterministic forecast.
While forecast systems are only really able to provide rainfall probabilities reliably to the nearest 10%,
different people have very different interpretations of expressions such as 'unlikely' (Patt; Schrag 2003),
so the use of numerical values, even where somewhat uncertain, is perhaps less ambiguous.


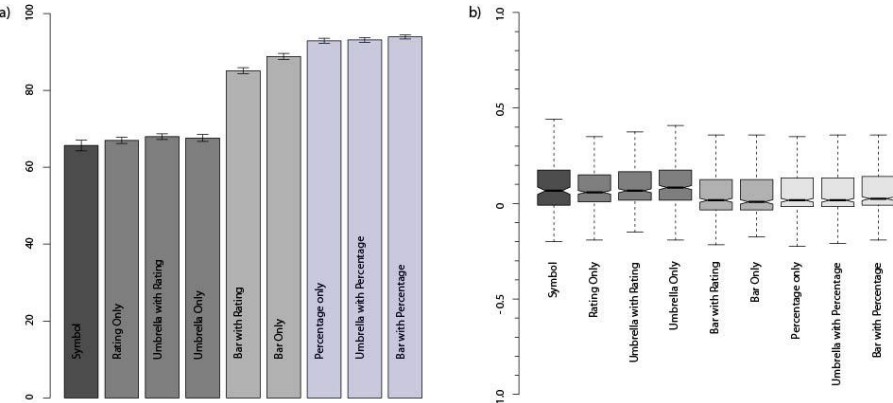



Figure 6: for each presentation format: a) mean of the percentage of questions each participant answers
correctly (error bars show standard error); b) mean difference between the actual and the participant's
specified probability (where notches on boxplots do not overlap there is significant difference between
the median values, positive values [negative values] represent an overestimation [underestimation] of the
actual probability.

The ability of participants to make the correct rainfall decision using different ways of presenting the
PoP forecast is shown in Fig. 6a. Fig. 6b shows the average difference between the actual probability and
the confidence specified by each participant for each presentation format. The best format would be one
with a median value close to zero, and a small range. Obviously we would not expect participants who
were presented with a symbol or only the probability rating to be able to provide precise estimates of the
actual probability, but the results for these formats can be used as a benchmark to determine whether
those presented with additional information content are able to utilise it.

Joslyn et al. (2009) find that using a pie graphic reduces reference class errors of PoP forecasts (although
not significant), and so it was hypothesised that providing a visual representation of uncertainty might
improve decision-making ability and allow participants to better interpret the probability.

For the first part of the rainfall question the best presentation formats are those where the percentage is
provided explicitly. The error bars overlap for these three formats so there is no definitive best format
identified from this analysis. Participants who were presented with 'Bar + Rating' or 'Bar Only' did not
perform as well, despite these presentation formats containing the same information. This suggests that





provision of the PoP as a percentage figure is vital for optimising decision-making. Note that participants
who were not presented with a Bar or Percentage would not have been able to answer all four questions
correctly without guessing.
For the second part of the rainfall question (Fig. 6b), there is no significant difference in the median
values for any of the formats that explicitly present the probability, the 'bar only' format is perhaps the
best due to the median being closer to zero. This result suggests that providing a good visual
representation of the probability is more helpful than the probability itself, though equally the bar may
just have been more intuitive within this game format for choosing the correct satellite button.
An interesting result, although not pertinent for presenting uncertainty, is that the median for those
participants who are only provided with deterministic information is significantly more than 0, and
therefore they are, on average, overestimating the chance of rain given the information. The
overestimation of probabilities for Q3 shifts 2 and 3, and Q4 Shift 1 (Fig. 5), where heavy rain showers
were forecast with chances of rain being 'high', shows that this may largely be to do with an
overestimation of the likelihood of rain when a rain symbol is included, though interestingly this is not
seen for the drizzle forecast in Q4 Shift 4, where all participants underestimate the chance of rain, or for
the light rain showers in Q1 Shift 1. This replicates the finding of Sivle (2014) which finds that some
people anticipate a large amount of rain to be a more certain forecast than a small amount of rain. Further
research could address how perceptions of uncertainty are influenced by the weather symbol, and if this
perception is well-informed (e.g. how often does rain occur when heavy rain showers are forecast).
### 4.3. What is the best presentation format for temperature forecasts?
The results for the different temperature presentation formats in each separate question (Fig. 4) are less
consistent than those for precipitation (Fig. 5), and the difference between estimated and actual
probabilities shows much more variability. It is expected that participants would find it more difficult to
infer the correct probability within the temperature questions, this is because they have to interpret the
probability rather than be provided it, as in the rainfall questions. The game was set up to mirror reality
in terms of weather forecast provision; rain / no rain is an easy choice for presentation of a probability,
but for summer temperature at least there is no equivalent threshold (arguably the probability of
temperature dropping below freezing is important in winter).
In Q4 around 70% of participants are able to make use of the extra level of information in Line 5090, but
in Q3 this extra uncertainty information appears to cause confusion compared to the more simplex
uncertainty representations. The difference in the responses between Q2 and Q3 is interesting; a 50%
correct result would be expected for the deterministic presentation formats because they have the same
forecast for the Saturday, so the outcomes highlight that participants are being influenced by some other
factor, perhaps the temperature on adjacent days.





Ignoring Line 50 90 because of this potential confusion, Fig. 7a suggests that Line 90 may be the best
presentation format for temperature forecasts. This would also be the conclusion for Fig. 7b, though a
smaller sample size within the deterministic formats means that the median value is not significantly
different from that for the Line presentation format. Like Tak, Toet and Erp (2015) an over optimistic
assessment of the likelihood of exceeding the temperature threshold has been found, with all presentation
formats overestimating the probability. However, the average of all the questions does not necessarily
provide a helpful indicator of the best presentation format because only four scenarios were tested, so the
results in Fig. 7 should be used with caution; the low standard errors reflect only the responses for the
questions that were provided.

The differences between the two different ways of presenting the deterministic information (Table and
Line), shown in Fig. 4 are of note because the UK Met Office currently provide forecasts in a more
tabular format. For Q2 and Q3 of the scenarios presented in this paper participants would be expected to
get the correct answer half of the time if they were only looking at the forecast values specific to the day
of interest (Saturday). The deviation of the responses from 50% shows that further work is needed to
address how people extract information from a presentation format. For example, Sivle (2015) finds
(from a small number of interviews) that informants were looking at weather symbols for the forecasts
adjacent to the time period they were interested in. While this study (and many others) have focussed on
the provision of information on weather forecast uncertainty it may be vital to also study differences in
interpretation of deterministic weather forecast presentation formats (from which a large proportion of
people infer uncertainty). This is also critical for putting in context the comparisons with presentation
formats that do provide uncertainty information. Fig. 4 shows that the differences between different
deterministic presentation formats are of the same magnitude as the differences between the deterministic
and probabilistic formats.

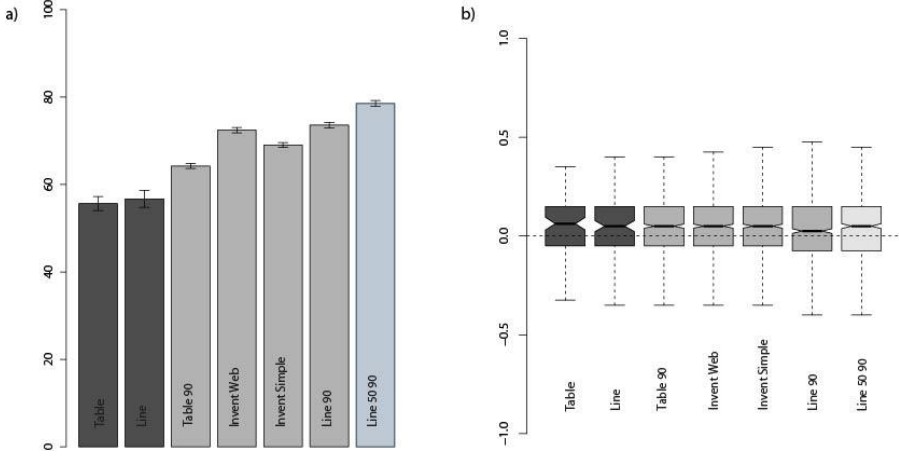



Figure 7: for each presentation format: a) mean of the percentage of questions each participant answers
correctly (error bars show standard error); b) mean difference between the actual and the participant's





specified probability (where notches on boxplots do not overlap there is significant difference between
the median values, positive values [negative values] represent an overestimation [underestimation] of the
actual probability.

### 4.4. How could the game be improved?

The main confounding factor within the results is how a particular weather scenario influenced a
participants' interpretation of the forecast (e.g. the drizzle result, or the influence of temperature forecasts
for adjacent days). The game could be improved by including a larger range of weather scenarios, perhaps
generated on-the-fly, to see how the type of weather influences interpretation. In practice this sounds
simple, but this is quite complex to code to take into account a plausible range of probabilities of rainfall
for each weather type (e.g. an 80% chance of rain is not likely for a 'sun' symbol), or that temperatures
are unlikely to reach a maximum of 0°C one day and 25°C the next (at least not in the UK).

The randomisation of the presentation format, week order and the outcome (based on the probability)
was significantly complex to code, so adding additional complexity without losing some elsewhere might
be unrealistic. Indeed, manually generating 16 realistic rainfall forecasts (4 weeks and 4 shifts); and 8
realistic temperature forecasts (4 weeks and 2 locations), and then the 9 (former) and 7 (latter)
presentation formats for each was difficult enough.

The game format is useful for achieving large numbers of participants, but perhaps understanding how
weather scenarios influence forecast interpretation is more appropriately studied through qualitative
methodologies such as that adopted by Sivle (2014), which was also able to find that weather symbols
were being interpreted differently to how the Norwegian national weather service intended.

### 4.5. How could this analysis be extended?

While not possible to break down the different presentation formats by socio-demographic influences, it
is possible using an ANOVA analysis to see where there are interactions between different variables. For
example, an ANOVA analysis for the mean error in rain confidence shows that there is no interaction
between the information content of the presentation format (e.g. deterministic, symbol, probability) and
the age or gender of the participant, but there is with their qualification (see Supplementary Material). A
full exploration of socio-demographic effects for both choice and confidence question types for rainfall
and temperature forecasts is beyond the scope of this paper, but we propose that further work could
address this and indeed the dataset is available to do so. However, we would note for those sceptical that
the provision of probabilistic forecasts would only lead to poorer decisions from those with lower
educational attainment, that while 86% of people who had attained a degree answered all four rainfall
questions correctly when presented with the probability only, 69% of those who had attained GCSE level
qualifications also answered all four questions correctly. In contrast, those with GCSE level
qualifications only got 15% of the questions right when presented with the weather symbol.




## 5. Conclusions

This study used an online game to build on the current literature and further our understanding of the ability of participants to make decisions using probabilistic rainfall and temperature forecasts presented in different ways and containing different complexities of probabilistic information. Employing an online game proved to be a useful format for both maximising participation in a research exercise and widening public engagement in uncertainty in weather forecasting.

Eosco (2010) states the necessity of considering visualisations as sitting within a larger context, and we followed that recommendation by isolating the presentation format from the potential influence of the television or web forecast platform where it exists. However, these results should be taken in the context of their online game setting – in reality the probability of precipitation and the temperature forecasts would likely be set alongside wider forecast information, and therefore it is conceivable that this might influence decision-making ability. Further, this study only accounts for those participants who are computer-literate, which might influence our results.

We find that participants provided with the probability of precipitation on average scored better than those without it, especially those who were presented with only the 'weather symbol' deterministic forecast. This demonstrates that most people provided with information on uncertainty are able to make use of this additional information. Adding a graphical presentation format alongside (a bar) did not appear to help or hinder the interpretation of the probability, though the bar formats without the numerical probability alongside aided decision-making, which is thought to be linked to the game design which asked participants to select a satellite button to state how sure they were that the rain / temperature threshold would be met.

In addition to improving decision making-ability, we found that providing this additional information on uncertainty alongside the deterministic forecast did not cause confusion when a decision could be made by using the deterministic information alone. Further, the results agreed with the findings of Joslyn and Savelli (2010), showing that people infer uncertainty in a deterministic weather forecast, and it therefore seems inappropriate for forecasters not to provide quantified information on uncertainty to the public.

*The Met Office started presenting the probability of precipitation on its website in late 2011. BBC Weather included it on their weather in 2018. The uncertainty in temperature forecast is not currently provided to the public by either of these websites.*

## 6. Data Availability

The game results are in the process of being uploaded to a repository and will be made freely available under license.




## 7. Author Contribution

All authors contributed to the design of the game. ES analysed the results and wrote the manuscript. All authors contributed comments to the final version of the manuscript.

## 8. Acknowledgements

This work was supported by the Industrial Mathematics KTN's Industrial Mathematics Internships Programme, co-funded by EPSRC and the Met Office. The authors would like to thank their colleagues at the Met Office for feedback on the design of the game, the technological development and the support in promoting the game to a wide audience.

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
