# Peer review of "The Met Office Weather Game: investigating how"

_Geoscience Communication, 2018_

## Referee Comment (RC1) · Skinner (Referee) · 10 Jan 2019

**Skinner (Referee)**

c.skinner@hull.ac.uk

Received and published: 10 January 2019

This paper highlights the results of a game-based survey into how people interpret probabilistic meteorological forecasts, focusing on temperature and likelihood of rain. The paper is well researched, written, and presented, and will be of interest to readers of Geoscientific Communication as it demonstrates how game-based methods can be used to engage the public with complex scientific information, and the best ways to present it so they can make decisions which are more likely to be advantageous. It is especially interesting as the methods of presentation surveyed have since been

adopted for public forecast communication.

The focus of the manuscript is very firmly on the survey results, their interpretation, and what they mean to the communication of probabilistic information. Details of the game itself – the theoretical framework, design, and development process – are limited but I appreciate that this isn't the scope of this manuscript. However, personally, I hope the authors do choose to share this in the future, possibly as supplementary material or further publication.

I have recommended that this paper be accepted with minor corrections -

Text in Figure 1 is small and cannot be read easily on a print out of the manuscript. Please consider a way of making this text larger (possibly flipping the image 90 degrees and filling the page).

In Section 2.1.3. consider expanding the description of the Briers Score for those might not be familiar with it – what does it measure, where is it used, and why it is an appropriate measure for this.

The authors use the term 'correct' to describe a user choice which is considered the probabilistically-speaking best option. This does not seem right to me – in a scenario where a user selects the best, or most sensible, option but gets rained on, I'm not sure a user would consider they had been 'correct but unlucky' (Line 313), but instead that it was the wrong choice. A user might instead opt for another location which is not such a probabilistically good choice maybe because they had a 'hunch' that the better option might rain, despite the odds. If their chosen location stays dry, then it was a 'correct' decision for them. This likely has little impact on this particularly survey, but if more complexity was added to the game such as each location having different footfall, the user begins to make decisions based on multiple criteria and the idea of one single choice being the correct one is not valid. The decision to interpret only one option as the correct one seems to be a deterministic way to interpret a probabilistic problem. I freely admit this is a trivial point on terminology but I'd be interested to hear the authors'

GCD
thoughts on it, and am happy for them to say I am wrong and move on.

Page 22, Lines 194 and 195 require a space between them.

Data Availability – Data ought to be available freely using an online repository with a DOI attached. If this is not possible, and it can only be made available under license, then details of how this can be obtained need to be included here before final publication.

Acknowledgements – Are there any project or grant codes which can be included here.

---

## Referee Comment (RC2) · Hut (Referee) · 14 Jan 2019

The authors have conducted a study where participants in an online game were presented with weather information in different formats and with different levels of information contained, to test how the information provided on the uncertainty in the weather forecast influences decision making by those receiving the weather forecast. I like the idea to use a game instead of classic survey or interview techniques as it forces people to actually make decisions instead of asking them what their decision would be.

I do have some concerns with the presentation and interpretations of the results that I would like to ask the authors and editor to consider before publishing the article in GC.

**1 On ensemble visualisation**

The authors focus their paper on the impact that providing uncertainty information has on decisions and they embed it within previous work by the atmospheric science community: nearly all their references come from this angle. The authors fail to acknowledge the complexities in he choice of how to visualize the outcome of an ensemble forecast in the first place. Given their scope (and, most likely, funding) I understand the choice to use the ensemble forecast visuals as they were made by the UK MET Office, but I would like to ask the authors to spend some words on explaining what the design decisions in this visualisation are based on. A quick google scholar search on "data visualization ensemble uncertainty" wields a host of papers from outside of geoscience with valuable insights, such as Obermaier 2014.

**2 On using a game as proxy for decision making**

As expressed above: I like the use of a game to gauge what decisions people would make based on uncertain information. However, I do believe that the scenario chosen to represent in the game can have influence on the outcome. By focussing on an icecream business scenario in summer I am afraid that the players in the game might be taking more risk than they would in real life. Even more so because, unlike in a real enterprise, there is no real cost associated with losing the game, which makes gameGCD
players take more risks than they would in real life1. This is an inherit problem with using games (and surveys for that matter), but I'd like to ask the authors to spend some more words on reflecting how the design choices in the "game" might have impacted the results. Could the risk taking nature of the game explain the offset in figure 4c where a median at zero is expected?

**3 On statistics**

The authors rightfully mention that in an online game it is relatively easy to get a large sample, more than 8.000 in their case. However, only using online users as the potential population to base your conclusions on could lead to skewed results when extrapolating to the entire general population. To get a first indication of how much the results of this study are applicable to the entire population, in figure 3 change the y axis to

Finally, the authors removed almost half of the sample because they had been taught about uncertainty. Since uncertainty is being treated in more and more disciplines, this limits the scope of the study to "can people who have not been taught uncertainty use forecasts that communicate uncertainty?". I do, however, believe that the group that has been taught uncertainty is of interest as well. By excluding them the authors make the (implicit) assumption that they would be better at interpreting uncertainty information in forecasts than others, something that I very much would like to see tested. I recommend the authors to either include this in the study, or to promise us a follow up study where those taught uncertainty are compared to those who did not receive that education. On improving the game The author state that "generating realistic forecasts was difficult enough". I would suggest not to let difficulty stop one from trying to further science. Given enough time (funding) one could envision that

GCD
<sup>1the reviewer would have shattered his legs jumping of cliffs on multiple occasions if he took the same risks in real life as in the games he plays...

weather forecasts be integrated in major online games and player actions could be tracked. Difficult, yes, but the "future work" section of an article is the place to dream big and I would invite the authors to do so.

Which also brings me to the author contributions. Did the authors work on the coding of the game itself? From the contributions it looks like they contributed to the design, but not the actual coding. Please have a look at the CRediT taxonomy to acknowledge all scientific roles that contributed to this work (https://casrai.org/credit/). (note to the editor: maybe campaign to have this journal support these roles? It has already been linked to ORCID profiles).

**4 On extending the analyses.**

The authors state looking deeper into different results for different age and educational levels is beyond the scope of the article (and I agree). They proceed, however, to still give a very quick comment on how results differ with educational attainment. They forego the care they take in presenting their other results in this quick remark though. I would ask the authors to either do the ANOVA analyses and present it as complete part of the study, or to remove the comment and invite the community to follow on this work using the data they will provide.

On a related note: I applaud the authors for making their data available. I do want to ask them to take extreme care in making the data anonymous enough to not be able to trace answers to individuals. Ask your local sociologists for advice on how to do this: our sociological colleagues have a long history of dealing with this.
**5 On presentation**

When printing the pdf on A4 paper the text in the figures was barely readable by this middle aged man. I ask the authors to expand the figures 4 and 5 into bigger separate figures.

**6 concluding**

I thoroughly enjoyed reading the paper and think it is of value to the geoscientific community. The above concerns can be addressed in minor revisions to the article.

H. Obermaier and K. I. Joy, "Future challenges for ensemble visualization," in-ÂăIEEE Computer Graphics and Applications, vol. 34, no. 3, pp. 8-11, 2014.Âă doi:10.1109/MCG.2014.52

---

## Short Comment (SC1) · 25 Jan 2019

This is a nice study that should be published pending addressing the reviewers' concerns. One issue that I noticed is that the authors should cite the following study that already looked at a group of UK students several years ago and asked them about their preferred ways of receiving forecasting information.

Peachey, J. A., D. M. Schultz, R. Morss, P. J. Roebber, and R. Wood, 2013: How forecasts expressing uncertainty are perceived by UK students. Weather, 68, 176–

181.

---

## Author Comment (AC1) · 25 Mar 2019

Many thanks for your constructive and considered comments. We will address your suggestions for minor corrections in the following ways:

Figure 1 text size - We will make some adjustments to improve it, and check in the proofs that it can be read easily.

Brier Score - we will adjust the text to read:

[Figure]

"Participants were scored (S) based on their specified confidence rating (C) and the outcome, using an adjustment of the Brier Score (BS) (see Table 1). The Brier Score was used as it measures the accuracy of probabilistic predictions. The use of the Brier Score means that if a participant is more confident they have more to gain, but also more to lose."

The use of 'correct' -

You have touched on the same debate we had when designing the game! However, our use of the word 'correct' is within the paper alone; the feedback to the participant within the game was a traffic light colour box with feedback along the lines of "You provided good advice, but on this occasion it rained", as per L319-320. We'll make this clearer on Line 312, by adjusting the text to

"we felt it important to provide some kind of feedback corresponding to whether they had accurately interpreted the forecast or not."

We accounted for the effect of the player adapting to the previous outcome / feedback by randomising the order in which each question appeared, and we also stored the data on this order so that it would be possible to see if there were learning effects. A preliminary analysis of the results didn't show any noteable learning effects across all the results, but it is possible that they are there for certain questions or for certain demographics.

Page 22 Lines 194 and 195 – thanks, we will adjust this.

Data availability: we are in the process of depositing the data

Acknowledgements: no, but we will acknowledge the consultancy that coded the game.

---

## Author Comment (AC3) · 25 Mar 2019

Thank you, we note that your paper is relevant and have updated the following sentence to include the citation:

"Morss et al. (2008), testing only non-graphical formats of presentation, found that the majority of people in a survey of the US public (n=1520) prefer a percentage (e.g. 10%) or non-numerical text over relative frequency (e.g. 1 in 10) or odds. For a smaller study of students within the UK (n=90) 90% of participants liked the probability format,

compared to only 33% for the relative frequency (Peachey et al., 2013). However, as noted by Morss et al. (2008), user preference does not necessarily equate with understanding."
* * *

---

## Author Response (AR1)

Response to reviewers

Reviewer 1:

Many thanks for your constructive and considered comments. We will address your suggestions for minor corrections in the following ways:

Figure 1 text size: We will make some adjustments to improve it, and check in the proofs that it can be read easily.

Add text to state why the Brier Score was used (get from David?)

The use of 'correct':

You have touched on the same debate we had when designing the game! However, our use of the word 'correct' is within the paper alone; the feedback to the participant within the game was a traffic light colour box with feedback along the lines of "You provided good advice, but on this occasion it rained", as per L319-320.  We'll make it clearer on Line 312, by adjusting the text to

"we felt it important to provide some kind of feedback corresponding to whether they had 'correctly' interpreted the forecast or not."

We accounted for the effect of the player adapting to the previous outcome / feedback by randomising the order in which each question appeared, and we also stored the data on this order so that it would be possible to see if there were learning effects. A preliminary analysis of the results didn't show any noteable learning effects across all the results, but it is possible that they are there for certain questions or for certain demographics.

Page 22 Lines 194 and 195 – thanks, we will adjust this.

Data availability: we are in the process of depositing the data

Acknowledgements: no, but we will acknowledge the consultancy that coded the game.

Reviewer 2:

Many thanks for your insightful comments which will improve the clarity and relevance of our manuscript. We respond to your numbered comments below.

1. Ensemble visualisation, reflect on why we went with particular choices.
   https://ieeexplore.ieee.org/abstract/document/6813969

Our choice for the visualisations and designs was based largely on a search of what visualisations were already in use by operational weather agencies at the time so that we were testing what was 'operational'. We already mention the 'Invent' format as coming from the Met Office website. The Line format was based on a format in use by the Norwegian weather service for their long term probability forecast (e.g. https://www.yr.no/place/Norway/Troms/Troms%C3%B8/Troms%C3%B8/long.html). The precipitation probability bar comes from the Australian Bureau of Meteorology website (e.g. http://www.bom.gov.au/nsw/forecasts/sydney.shtml).

We will update the text from Lines 164 onwards to read:

"The presentation formats used within this game were based on visualisations in use at the time by operational weather forecasting agencies. Seven different temperature forecast presentation formats were tested (Fig. 1), representing 3 levels of information content (deterministic, mean with 5th / 95th percentile range, mean with 5th / 95th and 25th / 75th. These included table and line presentation formats (in use by the Norwegian Weather Service, www.yr.no, for their long term probability forecast) as well as the 'Invent' style as it appeared on the web, and a more simplified version based on some user feedback. Nine different rainfall forecast presentation formats were tested (Fig. 2), with 3 different levels of information content including one deterministic format used as a control from which to draw comparisons. The 'bar format' is derived from the Australian Bureau of Meteorology website, www.bom.gov.au, and the 'umbrella' format was intended as a pictorial representation similar to a pie chart style found on the University of Washington's Probcast website (now defunct)"

2. A game as a proxy

The reviewer makes a good point here, we do understand that the game design cannot replicate the real world, and players may take more risks than in real life. We were trying to make a similar point towards the end of 4.4 but hopefully this updated text will make that clearer:

"The game format is useful for achieving large numbers of participants, but the game cannot replicate the real life costs of decision-making and therefore players might take more risks than they would in real life. While the aim was to compare different presentation formats it is possible that some formats encourage or discourage this risk taking more than others, especially if they need more time to interpret. A thorough understanding how weather scenarios influence forecast interpretation should be achieved by complementing game-based analysis such as this with qualitative methodologies such as that adopted by Sivle (2014), which was also able to find that weather symbols were being interpreted differently to how the Norwegian national weather service intended."

With respect to the bias, that could well reflect risk-taking but for 4c in particular it is perhaps more related to the construct of the question, with it only being possible to make an error in one direction when the probability is 0% (you can see the opposite for Q4 Stonemouth for example).

3. On statistics

The paragraph was cut off but we suspect the reviewer meant to finish this paragraph with a suggestion to change the plots in Figure 3 so that the y axis is a percentage rather than total. We are happy to do this.

The mentioning of the removal of those who have been taught about uncertainty within the text was a relic of an earlier version that was not caught on proof reading. We've changed the text and double checked all our figures / updated them where necessary. We will present the results shown in Figure 6 within the supplementary material for both responses to this 'taught about uncertainty' question.

The authors did not work directly on the coding itself, this was consulted out to an external company by the Met Office. We will include them in the acknowledgements.

4. Extending the analysis

The ANOVA analysis is presented in the Supplementary Material, where we have referenced this we have now included a direct reference to Figure 1 and the P Value of <2.2e-16.

We have only collected location data by Postcode District; there is a median population of over 20000 in each postcode district so identifying the individual would be difficult (see https://www.doogal.co.uk/PostcodeDistricts.php).

5. On presentation

There were some issues with how these plots could be presented within the Discussion version of the journal, we will make sure that they are readable in the proofs and if not adjust them accordingly.

6. Use of games in geoscience

We agree that more reference to games in geoscience is needed, within section 2 we will append a sentence to the paragraph that begins on Line 207:

"Our solution was to make the game as competitive as possible, while being able to identify and eliminate results from participants who played repeatedly to maximise their score. We also provided the incentive of the potential of a small prize to those that played all the way to the end of the game. Games have been used across the geosciences, for example to support drought decision-making (Hill et al., 2014), to promote understanding of climate change uncertainty (Pelt et al. 2015), and to test understanding of different visualisations of volcanic ash forecasts (Kelsey et al. 2017)."

Hill, H., Hadarits, M., Rieger, R., Strickert, G., Davies, E.G. and Strobbe, K.M., 2014. The Invitational Drought Tournament: What is it and why is it a useful tool for drought preparedness and adaptation?. *Weather and Climate Extremes*, *3*, pp.107-116.

Mulder, K.J., Lickiss, M., Harvey, N., Black, A., Charlton-Perez, A., Dacre, H. and McCloy, R., 2017. Visualizing volcanic ash forecasts: scientist and stakeholder decisions using different graphical representations and conflicting forecasts. Weather, Climate, and Society, 9(3), pp.333-348.

Van Pelt, S.C., Haasnoot, M., Arts, B., Ludwig, F., Swart, R. and Biesbroek, R., 2015. Communicating climate (change) uncertainties: simulation games as boundary objects. *Environmental science & policy*, *45*, pp.41-52.

Comment from David Schultz

Thank you, we note that the paper is relevant and have updated the following sentence to include the citation:

"Morss et al. (2008), testing only non-graphical formats of presentation, found that the majority of people in a survey of the US public (n=1520) prefer a percentage (e.g. 10%) or non-numerical text over relative frequency (e.g. 1 in 10) or odds. For a smaller study of students within the UK (n=90) 90% of participants liked the probability format, compared to only 33% for the relative frequency (Peachey et al., 2013). However, as noted by Morss et al. (2008), 
[revised manuscript text omitted]
user preference does not necessarily equate with understanding. For complex problems such as
communication of health statistics, research suggests that frequency is better understood than probability
(e.g. Gigerenzer et al. 2007), but for weather forecasts the converse has been found to be true, even when
a reference class (e.g. 9 out of 10 computer models predict that …) is included (Joslyn; Nichols 2009).
Joslyn and Nichols (2009) speculate that this response could be caused by the US public's long exposure
to the PoP forecast, or because weather situations do not lend themselves well to presentation using the
frequency approach because unlike for health risks they do not relate to some kind of population (e.g. 4
in 10 people at risk of heart disease).
As well as assessing the decision-making ability using a PoP forecast, it is also important to look at
potential methods for improving its communication. Joslyn et al. (2009) assess whether specifying the
probability of no rain or including visual representations of uncertainty (a bar and a pie icon) can improve
understanding. They found that including the chance of no rain significantly lowered the number of
individuals that made reference class errors. There was also some improvement when the pie icon was
added to the probability, which they suggested might subtly help to represent the chance of no rain. They
conclude that given the wide use of icons in the media more research and testing should be carried out
on the potential for visualisation as a tool for successful communication.
Tak, Toet and Erp (2015) considered public understanding of 7 different visual representations of
uncertainty in temperature forecasts among 140 participants. All of these representations were some form
of a line chart / fan chart. Participants were asked to estimate the probability of a temperature being
exceeded from different visualisations, using a slider on a continuous scale. They found systematic biases
in the data, with an optimistic interpretation of the weather forecast, but were not able to find a clear
'best' visualisation type.

**2.   Objectives and Methodology**

This study aims to address two concerns often vocalised by weather forecast providers about presenting
forecast uncertainties to the public; firstly, that the public do not understand uncertainty; and secondly,
that the information is too complex to communicate. Our aim was to build on the previous research of
Roulston and Kaplan (2009) and Roulston et al. (2006) by assessing the ability of a wider audience (not
only students) to make decisions when presented with probabilistic weather forecasts. Further, we aimed
to identify the most effective formats for communicating weather forecast uncertainty by testing different
visualisation methods and different complexities of uncertainty information contained within them (e.g.
a descriptive probability rating (Low (0%-20%), Medium (30%-60%) or High (70%-100%) compared to
the numerical value).
As such our objectives are as follows:

[revised manuscript text omitted]

the game cannot
replicate the real life costs of decision-making and therefore players might take more risks than they
would in real life. While the aim was to compare different presentation formats it is possible that some
formats encourage or discourage this risk taking more than others, especially if they need more time to
interpret. A thorough understanding how weather scenarios influence forecast interpretation should be
achieved by complementing game-based analysis such as this with qualitative methodologies such as
that adopted by Sivle (2014), which was also able to find that weather symbols were being interpreted
differently to how the Norwegian national weather service intended.

**4.5.  How could this analysis be extended?**

While not possible to break down the different presentation formats by socio-demographic influences, it
is possible using an ANOVA analysis to see where there are interactions between different variables. For
example, an ANOVA analysis for the mean error in rain confidence shows that there is no interaction
between the information content of the presentation format (e.g. deterministic, symbol, probability) and
the age or gender of the participant, but there is with their qualification (. P
value of $<2.2e^{-16}$, see Section 2 of the Supplementary Material). Initial analysis suggests subtle
differences between participants who have previously been taught or learnt about uncertainty compared
to those who have not (see Section 4, Supplementary Material), further analysis could explore this in
more detail at the level of individual questions.

A full exploration of socio-demographic effects for both choice and confidence question types for rainfall
and temperature forecasts is beyond the scope of this paper, but we propose that further work could
address this and indeed the dataset is available to do so. However, we would note for those sceptical that
the provision of probabilistic forecasts would only lead to poorer decisions from those with lower
educational attainment, that while 86% of people who had attained a degree answered all four rainfall
questions correctly when presented with the probability only, 69% of those who had attained GCSE level
qualifications also answered all four questions correctly. In contrast, those with GCSE level
qualifications only got 15% of the questions right when presented with the weather symbol.

**5.  Conclusions**

This study used an online game to build on the current literature and further our understanding of the
ability of participants to make decisions using probabilistic rainfall and temperature forecasts presented
in different ways and containing different complexities of probabilistic information. Employing an online
game proved to be a useful format for both maximising participation in a research exercise and widening
public engagement in uncertainty in weather forecasting.
Eosco (2010) states the necessity of considering visualisations as sitting within a larger context, and we
followed that recommendation by isolating the presentation format from the potential influence of the
television or web forecast platform where it exists. However, these results should be taken in the context
of their online game setting – in reality the probability of precipitation and the temperature forecasts
would likely be set alongside wider forecast information, and therefore it is conceivable that this might
influence decision-making ability. Further, this study only accounts for those participants who are
computer-literate, which might influence our results.
We find that participants provided with the probability of precipitation on average scored better than
those without it, especially those who were presented with only the 'weather symbol' deterministic
forecast. This demonstrates that most people provided with information on uncertainty are able to make
use of this additional information. Adding a graphical presentation format alongside (a bar) did not appear
to help or hinder the interpretation of the probability, though the bar formats without the numerical
probability alongside aided decision-making, which is thought could be linked to the game design
which asked participants to select a satellite button to state how sure they were that the rain / temperature
threshold would be met.
In addition to improving decision making-ability, we found that providing this additional information on
uncertainty alongside the deterministic forecast did not cause confusion when a decision could be made
by using the deterministic information alone. Further, the results agreed with the findings of Joslyn and
Savelli (2010), showing that people infer uncertainty in a deterministic weather forecast, and it therefore
seems inappropriate for forecasters not to provide quantified information on uncertainty to the public.
*The Met Office started presenting the probability of precipitation on its website in late 2011. BBC*
*Weather included it on their weather in 2018. The uncertainty in temperature forecast is not currently*
*provided to the public by either of these websites.*

**6. Data Availability**

The dataset
analysed within this paper is available under license from http://dx.doi.org/10.17864/1947.198.

**7. Author Contribution**

All authors contributed to the design of the game. ES analysed the results and wrote the manuscript. All authors contributed comments to the final version of the manuscript.

**8. Acknowledgements**

This work was supported by the Industrial Mathematics KTN's Industrial Mathematics Internships Programme,This project (IP10-022) was part of the programme of industrial mathematics internships managed by the Knowledge Transfer Network (KTN) for Industrial Mathematics, and was co-funded by EPSRC and the Met Office. The authors would like to thank their colleagues at the Met Office for feedback on the design of the game, the technological development and the support in promoting the game to a wide audience. Software design was delivered by the digital consultancy 'TwoFour'.